MetaSwin: a unified meta vision transformer model for medical image segmentation

Lee Soyeon 1
Lee Minhyeok mlee@cau.ac.kr 2
1 Department of Intelligent Semiconductor Engineering, Chung-Ang University , Seoul , South Korea
2 School of Electrical and Electronics Engineering, Chung-Ang University , Seoul , South Korea
Wan Shibiao
Electronic publication date: 2024 Jan 3
Publication date: 2024
Volume: 10
Electronic Location ID: e1762
Received 2023 Sep 6; Accepted 2023 Nov 27
Copyright: ©2024 Lee and Lee
Copyright year: 2024
Copyright holder: Lee and Lee
License: This is an open access article distributed under the terms of the Creative Commons Attribution License, which permits unrestricted use, distribution, reproduction and adaptation in any medium and for any purpose provided that it is properly attributed. For attribution, the original author(s), title, publication source (PeerJ Computer Science) and either DOI or URL of the article must be cited.
License URL: https://creativecommons.org/licenses/by/4.0/

Keywords: Medical image segmentation, Transformer, Spatial pooling, Vision transformers, MRI

Funding: The National Research Foundation of Korea (NRF) grant funded by the Korea government (MSIT) RS-2023-00251528 This research was supported by the National Research Foundation of Korea (NRF) grant funded by the Korea government (MSIT) (No. RS-2023-00251528). The funders had no role in study design, data collection and analysis, decision to publish, or preparation of the manuscript.

==============================
Transformers have demonstrated significant promise for computer vision tasks. Particularly noteworthy is SwinUNETR, a model that employs vision transformers, which has made remarkable advancements in improving the process of segmenting medical images. Nevertheless, the efficacy of training process of SwinUNETR has been constrained by an extended training duration, a limitation primarily attributable to the integration of the attention mechanism within the architecture. In this article, to address this limitation, we introduce a novel framework, called the MetaSwin model. Drawing inspiration from the MetaFormer concept that uses other token mix operations, we propose a transformative modification by substituting attention-based components within SwinUNETR with a straightforward yet impactful spatial pooling operation. Additionally, we incorporate of Squeeze-and-Excitation (SE) blocks after each MetaSwin block of the encoder and into the decoder, which aims at segmentation performance. We evaluate our proposed MetaSwin model on two distinct medical datasets, namely BraTS 2023 and MICCAI 2015 BTCV, and conduct a comprehensive comparison with the two baselines, i.e., SwinUNETR and SwinUNETR+SE models. Our results emphasize the effectiveness of MetaSwin, showcasing its competitive edge against the baselines, utilizing a simple pooling operation and efficient SE blocks. MetaSwin’s consistent and superior performance on the BTCV dataset, in comparison to SwinUNETR, is particularly significant. For instance, with a model size of 24, MetaSwin outperforms SwinUNETR’s 76.58% Dice score using fewer parameters (15,407,384 vs 15,703,304) and a substantially reduced training time (300 vs 467 mins), achieving an improved Dice score of 79.12%. This research highlights the essential contribution of a simplified transformer framework, incorporating basic elements such as pooling and SE blocks, thus emphasizing their potential to guide the progression of medical segmentation models, without relying on complex attention-based mechanisms.

Introduction

Deep learning has irrevocably impacted medical imaging, where deep learning-based applications have transcended traditional techniques in terms of efficiency, accuracy, and diagnostic quality. These applications range from organ delineation to lesion localization and diagnostic precision on a variety of imaging modalities such as computed tomography (CT) and magnetic resonance imaging (MRI). Medical image segmentation techniques are useful to assist in diagnosing the patient and establishing an intervention plan (Valanarasu & Patel, 2022; Wang, Huang & Yang, 2023). In addition to diagnosis and medical plans, delineated tumor analysis can be applied to examining the progression of the malignant object and predicting life expectancy (Joshua et al., 2022). The pervasive influence of deep learning-based technology in medical imaging calls for a continuous effort to refine and further optimize these applications to benefit clinical decision-making processes.

AI-based medical image segmentation techniques have shown prominence in providing precise and efficient results. Recently, transformer-based models have gained significant attention and achieved innovative performance in various computer vision fields (Tabik et al., 2020; Tang et al., 2020; Zeid, El-Bahnasy & Abo-Youssef, 2021). Since the groundbreaking research of Vision Transformer (ViT) (Dosovitskiy et al., 2020), which adapts pure transformer to image classification tasks, demonstrated that transformers could be employed beyond natural language processing, it stimulates a surge of research activity in computer vision. To make further progress and achieve promising performance in medical image segmentation as well, various follow-up models are generated (Ambita, Boquio & Naval, 2021; Tyagi et al., 2021).

Following the success of ViT, the Swin Transformer (Liu et al., 2021) was introduced, featuring a hierarchical structure that enabled the examination of varying scales via the Shifted Window concept. This design demonstrated its versatility and adaptability by providing a path to higher performance in numerous downstream tasks. Building on this innovation, a new model known as SwinUNETR (Hatamizadeh et al., 2022a) was created by making use of a U-shaped network with a Swin transformer as the encoder and links it to a CNN-based decoder using skip connections at various resolutions. This model demonstrated particularly improved performance in semantic segmentation of medical images.

However, recent studies suggest that the performance of the transformer-based model is more dependent on the general architecture MetaFormer than on a particular attention module (Yu et al., 2022). Transformer based-models can maintain considerably high performance with relatively low computational complexity even when the attention-based module is switched out for the extremely simple non-parametric operator pooling, which basically operates the most fundamental token mixing. However, SwinUNETR encounters the issue of prolonged training and inference runtimes, a problem specifically caused by the incorporation of the attention mechanism within the architecture. Additionally, PoolFormer is primarily designed as an improved model within the Transformer structure and is not tailored specifically for U-shaped architectures used in semantic segmentation.

Inspired by PoolFormer, which replaces the attention module in Transformers with a spatial pooling operator, we propose MetaSwin, a new modified U-shaped architecture that incorporates PoolFormer. Our study builds upon the established SwinUNETR model, well-recognized for its efficacy in 3D image segmentation. Therefore, MetaSwin integrates the concepts from PoolFormer into SwinUNETR by substituting attention-based modules in SwinUNETR with an extremely simple spatial pooling operator. In addition to this fundamental transformation, we also include Squeeze-and-Excitation (SE) blocks after each MetaSwin block of the encoder and to the decoder, adapted from SE Networks (Hu, Shen & Sun, 2018), which have been successful in recalibrating channel-wise feature responses, promising to enhance the model’s accuracy. We validate the effectiveness and efficiency of our model with two distinct medical datasets: the BraTS 2023 segmentation challenge dataset, which focuses on brain tumor segmentation, and the BTCV dataset, which contains abdominal imaging data. By comparing the performance of MetaSwin on these datasets to the traditional transformer-based model, SwinUNETR, we aim to demonstrate the potential of MetaSwin as a robust and efficient tool in 3D image semantic segmentation alongside a significant reduction in computational complexity and training duration.

Related Work

Medical segmentation deep learning models

The U-Net (Ronneberger, Fischer & Brox, 2015) architecture has proven its efficacy in medical image segmentation, attracting widespread adoption in both CNN-based (Çiçek et al., 2016; Futrega et al., 2021; Huang et al., 2020; Isensee et al., 2020; Jin et al., 2019; Luu & Park, 2022; Myronenko, 2019; Zhou et al., 2020) and more recent Transformer-based approaches (Peiris et al., 2021; Petit et al., 2021; Wang et al., 2021; Xie et al., 2021; Zheng et al., 2021). The U-shaped design comprises an encoder and decoder. The encoder progressively generates lower-resolution representations imbued with heightened semantic significance and encompassing global contextual insights. Subsequently, the decoder reconstructs a high-resolution segmented image.

Jin et al. (2019) introduced a refinement termed DUNet, wherein each 3  × 3 convolutional layer of the original U-Net was substituted with a deformable convolutional block. This adaptation, tailored for precise retinal vessel segmentation, accommodates intricate vessel structures by dynamically adjusting receptive fields to the varied forms and sizes present in input features.

Huang et al. (2020) extended the U-Net paradigm with U-Net3+, which enhances dense skip connections from U-Net++ by incorporating comprehensive full-scale skip connections. These linkages forge connections between each decoder level and every preceding encoder level, thus fostering robust information exchange.

The triumph of transformer models in natural language processing (NLP) inspired their transition into the realm of visual recognition. Dosovitskiy et al. (2020) introduced the ViT, which can be used to extract fine-grained features directly from sequences of image patches. They divide an image into patches and provide a Transformer the linear embeddings of these patches in sequence. Image patches are considered in the same manner as tokens (words) in an NLP. ViT resolves the long-range dependency between images by applying global attention to 16 × 16 patches of the entire image and focusing on its global significant features. They demonstrate that a ViT pre-trained on an extensive proprietary dataset can accomplish outstanding results on supervised image classification tasks.

Chen et al. (2021) pioneered the fusion of ViT and U-Net in an approach to tackle the challenge of long-range dependencies. By harnessing ViT’s features from distinct stages, UNETR (Hatamizadeh et al., 2022b) emerged as an encoder–decoder framework. It employs ViT as the encoder, generating hierarchical feature maps that cascade into the decoder, culminating in the creation of precise segmentation masks. This synthesis capitalizes on the innate strengths of ViT’s global context extraction in the bottleneck block, while the encoder and decoder pivot toward local feature retrieval using localized self-attention layers.

The Swin Transformer (Liu et al., 2021), known for its ability to handle hierarchical structures, has become a key component in vision-related tasks. Its defining feature is the shifting window technique, which systematically arranges sequential window shifts to capture localized context through patch-level analysis. The Swin Transformer separates the patches into non-overlapping windows and restricts the self-attention computation in the limited or shifted windows, which increases localized inductive bias. Additionally, it makes use of patch-merging layers to create hierarchical models that are effective for downstream tasks such as object detection and semantic segmentation. This design, combined with the successful U-Net-inspired architecture, forms the basis for SwinUNETR, a superior model in the field of medical segmentation.

SwinUNETR (Hatamizadeh et al., 2022a), specifically designed for the semantic segmentation of brain tumors in multi-modal MRI data, presents a unique structure. The Swin Transformer functions as the encoder, while a CNN-based decoder forms connections through skip connections at different resolutions. This collaboration ensures a comprehensive information flow, leading to improved segmentation accuracy. The integration of the Swin Transformer’s robust hierarchical features with a U-Net-like framework represents the core innovation of SwinUNETR’s design.

MetaFormer

The application of transformers in computer vision has shown remarkable potential. The prevailing consensus attributes much of their efficacy to the attention-based token mixing component. Recent investigations have brought to light the possibility of substituting the attention-based module of transformers with spatial MLPs, resulting in models that retain commendable performance (Tolstikhin et al., 2021; Touvron et al., 2022). Through their empirical analysis, it has been posited that a model’s effectiveness is influenced more by the overall architecture of transformers rather than the specific token mixing module employed. The research focuses on determining the performance lower limit and adaptability of the model by leveraging common operators rather than developing innovative token mixing methods.

A novel conceptual framework known as MetaFormer (Yu et al., 2022) has emerged, drawing inspiration from the transformer paradigm while abstracting the intricacies of token mixing approaches. By omitting the token mixer, MetaFormer stands as a generalized architectural construct derived from transformers. A deliberate replacement of Transformers’ attention module with a straightforward spatial pooling operator is undertaken to probe the core token mixing aspects. Termed as PoolFormer, this approach significantly reduces computational complexity compared to transformer-based networks that employ multi-head self-attention. Intriguingly, PoolFormer showcases competitive performance across an array of computer vision tasks. This emphasizes the need for future research projects aimed towards improving MetaFormer rather than concentrating only on token mixer modules. The suggested PoolFormer also provides a starting point for future architectural improvements within the MetaFormer framework.

Materials & Methods

We propose a novel design called MetaSwin by adapting SwinUNETR and incorporating ideas from MetaFormer. Furthermore, we additionally include SE blocks in our architecture to allow our model to dynamically prioritize important information. This model addresses the shortcomings of SwinUNETR, notably its resource-intensive computations and prolonged training/inference times. We showcase the proficiency and potency of MetaSwin across diverse datasets for segmentation tasks, achieving competitive performance compared to the latest benchmarks.

U-shaped architecture

In our study, we introduce a U-shaped architecture for semantic segmentation, which is conventionally used in this domain. Based on this conventional architecture, we propose a new modified U-Net design. This design draws inspiration from the SwinUNETR model, effectively merging the capabilities of two influential models, Swin Transformer and U-Net. The architecture takes an input image slice and processes it through an encoder–decoder framework.

The encoder part of our model is based on MetaSwin blocks, where we substitute the attention-based modules in Swin Transformer with a much simpler average pooling operator. Each MetaSwin block can be then followed by a SE block, which helps in recalibrating channel-wise feature responses, thereby enhancing the model’s accuracy.

On the other hand, the decoder consists of ResNet-based (He et al., 2016) blocks, and it is connected to the encoder through skip connections. The decoder also incorporates residual blocks of various resolutions, and each of these blocks is complemented by its own SE block.

MetaSwin block

Recently, attention mechanisms have gained significant attention in the deep learning community, and many efforts have been made to design attention-based token mixer components (D’Ascoli et al., 2021; Han et al., 2021). The Swin Transformer model, known for its attention-based transformer architecture and the clever utilization of shifted windows technique, has been successful in various tasks.

However, recent researches, particularly the MetaFormer study (Yu et al., 2022), has highlighted that the general architecture MetaFormer, not the equipped particular token mixers such as the attention module, is the primary source of the competency of transformer/MLP like models. This insight suggests that even a straightforward pooling operator can perform effectively in place of the attention-based token mixer.

Motivated by the findings of MetaFormer, we propose our novel MetaSwin model in this work. MetaSwin is comprised of multiple stages, each containing a MetaSwin Block. As depicted in Fig. 1, within each MetaSwin Block, we replace the traditional Multi-head self-attention module of Swin Transformer with a much simpler average pooling operator.

Figure 1 The overall framework of MetaSwin+SE.

MetaSwin+SE is the addition of SE blocks to MetaSwin. The model uses a 4-stage hierarchical architecture. Each stage contains a MetaSwin Block and SE Block as depicted in the figure. The multi-head self-attention module of the traditional transformer model is replaced by the average pooling operator within each MetaSwin Block. SE Block is applied after each MetaSwin Block in the encoder and each residual block in the decoder. MetaSwin is the model without any SE blocks.

The key advantage of this substitution lies in the computational complexity. While self-attention typically demands a quadratic amount of computation with respect to the tokens being mixed, the pooling operator only requires linear complexity, making it computationally efficient. Moreover, the pooling operator involves no learnable parameters, as it merely averages the attributes of each token’s neighboring tokens.

The pooling operator can be represented as T:i,j′=1K×K∑p,q=1KT:,i+p−K+12,j+q−K+12,

where K denotes the kernel size.

In stage 1 of the MetaSwin block, we employ a linear embedding layer to create 3D tokens with reduced dimensions H2×W2×D2 and set them in a dimension C embedding space. Subsequently, at the end of each stage, the resolution of the feature representations is halved to preserve the hierarchical structure of the encoder.

To summarize, the MetaSwin model comprises four stages, each containing MetaSwin Blocks with resolutions of H2×W2×D2, H4×W4×D4, H8×W8×D8, and H16×W16×D16, respectively. This design choice allows our model to efficiently process and extract meaningful features from the input image slice, leading to promising results in semantic segmentation tasks.

Squeeze-and-excitation (SE) block

In order to enable our model to dynamically prioritize essential information, we introduce SE blocks into our architecture, which is applied after each MetaSwin block after each MetaSwin block of the encoder and to the decoder., as illustrated in Fig. 1.

SE blocks employ global average pooling on the feature maps to capture global information and create a compact channel descriptor. This descriptor is then processed through two fully connected layers to generate channel-wise scaling factors. These adaptive weights allow the model to emphasize important channels while downplaying less relevant ones. As a result, the recalibration mechanism enhances the model’s discriminative power, enabling it to focus on the most informative features and improve semantic segmentation performance. Thus, we provide our model with the flexibility to use the SE block to selectively emphasize important features, which can lead the improvement of the result of segmentation tasks.

Datasets

To evaluate the performance of our proposed MetaSwin model, we conduct experiments on two distinct medical imaging datasets. The first dataset is provided by the BraTS challenge, which offers a collection of 3D MRI scans with voxel-wise ground truth labels for brain tumor segmentation. This dataset, BraTS 2023 (Baid et al., 2021; Bakas et al., 2017; Menze et al., 2014), comprises 5,880 MRI scans from 1,470 subjects, each containing four 3D MRI modalities: native (T1), post-contrast T1-weighted (T1Gd), T2-weighted (T2), and T2 fluid attenuated inversion recovery (T2-FLAIR) volumes. These scans were acquired using diverse clinical protocols and scanners from multiple data-contributing institutions. The input image size is set to 240 × 240 × 155, and the dataset includes four distinct classes: background (label 0), necrotic/non-enhancing tumor (label 1), peritumoral edematous/invaded tissue (ED) (label 2), and GD-enhancing tumor (label 4). Due to the unavailability of publicly accessible testing sets, we conduct our experiments using five-fold cross-validation, with 80% of the data used for training and the remaining 20% for validation. The training set consists of 1,251 samples, while the validation set includes 219 samples.

The second dataset, the 2015 MICCAI Multi-Atlas Labeling Beyond the Cranial Vault (BTCV) Challenge dataset, contains 30 portal venous contrast phase CT images with manual labels for 13 abdominal organs (adrenal, aorta, esophagus, gallbladder, kidney, liver, pancreas, spleen and portal vein, spleen, stomach, and vena cava). This dataset is divided into 30 training subjects and 20 test subjects. We split the training dataset into an 80:20 ratio for training and validation purposes, respectively. The training set consists of 2,212 axial slices, totaling 3,779 axial contrast-enhanced abdomen CT images. Each slice has 512 × 512 pixels and the in-plane resolutions range from 2.5 mm to 5.0 mm, while the slice thicknesses range from 0.54 × 0.54 mm2 to 0.98 × 0.98 mm2.

Implementation details and evaluation methods

For the implementation of MetaSwin, we utilize PyTorch and MONAI (Cardoso et al., 2022) frameworks, running the experiments on NVIDIA RTX A6000 GPUs. The training process was carried out with a batch size of 1 per GPU and a learning rate of 0.0001. For the BraTS dataset, all models were trained for 800 epochs and the input images were cropped into randomized patches of size 128 × 128 × 128. The embedding space C is set to 48 in stage 1 for all models. For the BTCV dataset, we employ 6 different feature sizes for each model to compare the performance of models in various sizes by varying the dimensions of embedded space C: 24, 36, 48, 60, 72, and 84. The training was conducted for 2,000 epochs, and the input resolution was set to 96 × 96.

As the loss function, we employ the Soft Dice loss, calculated voxel-wise with the formula DiceLoss=1−2∑i=1npigi ∑i=1npi2+ ∑i=1ngi2,

where pi and gi are corresponding pixel values for prediction and ground truth and n is the total number of pixels in the image.

Results

Segmentation results with the BraTS 2023 dataset

We conduct a set cross-validation split and evaluated each model’s performance across all five folds using the BraTS 2023 dataset, resulting in a total of 20 experiments across four models. We compare the performance of MetaSwin to traditional SwinUNETR and SwinUNETR+SE, which includes SE blocks added to the encoder and decoder of SwinUNETR in a manner similar to MetaSwin.

Table 1 shows the results of each model across all five folds on the BraTS 2023 dataset. Surprisingly, despite simple pooling operator replacing attention only, MetaSwin can still obtain extremely competitive performance compared to the winning methologies of past years. While there are instances where MetaSwin exhibits slightly lower average Dice scores under certain conditions compared to the baseline models, it is essential to emphasize overall performance of MetaSwin remains highly competitive, accompanied by substantial reductions in computational complexity and training time. On average, the proposed MetaSwin outperforms SwinUNETR and separately, MetaSwin obtains competitive performance in most of sub-regions classes and five folds. Specifically, MetaSwin performed the highest average Dice score in Fold 2. Additionally, MetaSwin requires 1.93% and 2.62% fewer parameters than SwinUNETR and SwinUNETR+SE, respectively, resulting in reduced learning time by 9.04% and 18.58%, respectively. Furthermore, MetaSwin+SE, where SE blocks are added into MetaSwin, still achieves performance that is competitive with fewer parameters and reduced learning time by 1.92% and 15.61%, compared to SwinUNETR+SE, respectively.

Table 1 Comparison of Five-fold cross-validation performance in SwinUNETR, SwinUNETR+SE, MetaSwin and MetaSwin+SE on BraTS 2023 dataset.

winUNETR+SE and MetaSwin+SE are the addition of SE blocks to SwinUNETR and MetaSwin each.

Model	Dice score	Fold 1	Fold 2	Fold 3	Fold 4	Fold 5	Avg.	Num. of parameters	Training time (mins)	
SwinUNETR	TC	0.51970	0.52240	0.53650	0.54916	0.52050	0.52965	62,191,941	20,426	
WT	0.70440	0.74230	0.73480	0.72261	0.72200	0.72522	
ET	0.39720	0.41960	0.43110	0.42231	0.40650	0.41534	
Avg.	0.54043	0.56143	0.56747	0.56469	0.54967	0.55674	
SwinUNETR+SE	TC	0.53037	0.53450	0.55048	0.53627	0.54123	0.53857	62,613,669	22,213	
WT	0.72526	0.74288	0.73920	0.72930	0.73946	0.73522	
ET	0.40685	0.42487	0.44479	0.41512	0.42255	0.42284	
Avg.	0.55416	0.56742	0.57816	0.56023	0.56775	0.56554	
MetaSwin	TC	0.51452	0.53941	0.52152	0.54020	0.52777	0.52868	61,012,581	18,733	
WT	0.71977	0.74849	0.73178	0.73005	0.74035	0.73409	
ET	0.38848	0.41930	0.41831	0.41084	0.40217	0.40782	
Avg.	0.54387	0.56906	0.55769	0.56113	0.55873	0.55810	
MetaSwin+SE	TC	0.52297	0.52640	0.52500	0.54888	0.53110	0.53087	61,434,309	19,213	
WT	0.72096	0.74577	0.72730	0.72501	0.73120	0.73005	
ET	0.40262	0.42449	0.42330	0.42437	0.40970	0.41690	
Avg.	0.54885	0.56555	0.55853	0.56609	0.55733	0.55927	
Notes.

ET Enhanced Tumor

TC Tumor Core

WT Whole Tumor

Bold values indicate the best performance.

The utilization of the pooling operator enables each token to uniformly gather features from its neighboring tokens, making it a remarkably simple token-mixing process. Moreover, the inclusion of the SE block, a powerful yet straightforward mechanism, performs a significant role in capturing global spatial information and creating an activation vector to rescale the feature maps, thereby emphasizing more critical information.

In Fig. 2, the values represent the average results of the 5-fold cross-validation, indicating a single data point per model. Figure 2 demonstrates explicitly MetaSwin surpasses SwinUNETR with fewer parameters and less training time. In comparison to SwinUNETR, not just MetaSwin but also MetaSwin+SE demonstrates competitive results at lower computational costs. Analyzing each model separately reveals that models with SE addition perform better with more training parameters and time than models without SE blocks. These findings imply the significance of a general transformer architecture with relatively fundamental operators, such as pooling or an SE block, in developing vision models, surpassing the reliance on designing complex attention-based transformer models.

Figure 2 Average validation Dice score on BraTS 2023 in terms of the number of parameter and training time (mins).

SwinUNETR+SE and MetaSwin+SE are the addition of SE blocks to SwinUNETR and MetaSwin each.

Figure 3 presents qualitative segmentation comparisons for brain tumors. The segmentation outputs for the three sub-regions are clearly defined and consistent with quantitative results. Rows 2 and 3 in Fig. 3 illustrate our model’s improved ability to capture intricate tumor characteristics.

Segmentation results with BTCV dataset

Table 2 shows the performance of various model sizes on the BTCV dataset. We assessed model performance across varying feature sizes by adjusting the dimensions of the embedding space C. As we compared six sizes for each of the four models, this led to a total of 24 experiments. Remarkably, the proposed MetaSwin consistently achieves highly competitive performance compared to SwinUNETR in terms of Test Dice, despite occasional instances where MetaSwin exhibits slightly lower scores under specific conditions. This highlights the overall competitiveness of MetaSwin while also delivering substantial reductions in computational complexity and training time. For instance, specifically in case of a model size of 24, MetaSwin still shows better performance when SwinUNETR obtains 76.58% Dice score with 15,703,304 parameters and 467 mins training time while MetaSwin reaches 79.12% Dice score with 1.92% fewer parameters (15,407,384) and 55.6% fewer training time (300) than those of SwinUNETR. Additionally, MetaSwin produced competitive test Dice score results compared to SwinUNETR+SE where SE blocks are added into SwinUNETR. For example, in a large model of size 84, MetaSwin outperforms SwinUNETR+SE with fewer computational costs by significant amounts. This finding suggests a great potential of the general structure of transformer, i.e., MetaFormer as compared to the attention-based transformer in segmentation tasks.

Figure 3 Qualitative comparisons for brain tumor segmentation on BraTS 2023 between GT, SwinUNETR, MetaSwin and MetaSwin+SE.

Our models accurately capture the detailed information in segmentation outputs. The combination of the red, blue, and green regions creates the whole tumor (WT). The tumor core (TC) is formed by merging the red and blue areas, while the blue regions represent the enhancing tumor core (ET).

Table 2 Comparison with SwinUNETR, SwinUNETR+SE, MetaSwin, and MetaSwin+SE on BTCV dataset.

SwinUNETR+SE and MetaSwin+SE are the addition of SE blocks to SwinUNETR and MetaSwin each. Size means the size of the embedding space C in stage 1 of the encoder.

Model	Size	Num. of parameters	Training time (mins)	Max. Val Acc	Test DICE	
SwinUNETR	24	15,703,304	467	0.82016	76.58%	
36	35,072,996	649	0.83332	80.17%	
48	62,187,296	761	0.83596	79.90%	
60	97,046,204	942	0.83250	80.37%	
72	139,649,720	1,144	0.83408	79.74%	
84	189,997,844	1,334	0.83823	80.51%	
SwinUNETR+SE	24	15,809,336	477	0.81962	77.15%	
36	35,310,668	668	0.82865	80.62%	
48	62,609,024	783	0.83709	79.50%	
60	97,704,404	972	0.83729	81.12%	
72	140,596,808	1,176	0.83546	81.20%	
84	191,286,236	1,373	0.84152	81.02%	
MetaSwin	24	15,407,384	300	0.81549	79.12%	
36	34,408,796	484	0.82300	79.33%	
48	61,007,936	601	0.83581	80.94%	
60	95,204,804	773	0.83967	81.07%	
72	136,999,400	970	0.84168	80.84%	
84	186,391,724	1,167	0.84192	81.36%	
MetaSwin+SE	24	15,513,416	313	0.81738	76.76%	
36	34,646,468	499	0.83117	80.17%	
48	61,429,664	612	0.83003	81.18%	
60	95,863,004	798	0.84058	79.48%	
72	137,946,488	1,001	0.83994	81.16%	
84	187,680,116	1,196	0.84179	81.00%	
Notes.

Bold values indicate the best performance.

Furthermore, considering the addition of SE blocks model, even in a large model of size 84 for instance, MetaSwin+SE, in which SE blocks are added into MetaSwin, is paramount to SwinUNETR+SE with fewer parameters and training time by 1.91% and 14.8%.

Figure 4 clearly shows that MetaSwin and MetaSwin+SE are paramount to SwinUNETR with fewer parameters and training time. These findings further validate the effectiveness and practicality of our proposed MetaSwin model. However, it is important to note that Fig. 4 also reveals a noticeable drop in the test Dice scores for MetaSwin+SE and SwinUNETR+SE. This phenomenon may be attributed to the interaction between the introduced SE blocks and models of specific sizes. It is plausible that the integration of SE blocks may not align as effectively with the distinctive design principles and characteristics of MetaSwin and SwinUNETR, potentially resulting in a negative impact on their performance.

Figure 4 Test Dice score on BTCV dataset in terms of the number of parameters and training time (mins).

SwinUNETR+SE and MetaSwin+SE are the addition of SE blocks to SwinUNETR and MetaSwin each.

Conclusions

In this study, we introduce an innovative and practical approach, the MetaSwin model, within the domain of deep learning for semantic segmentation tasks. Through the strategic replacement of attention-based modules within the SwinUNETR framework with a straightforward spatial pooling operator, our work has effectively unveiled a more streamlined and efficient model architecture. Moreover, the incorporation of SE blocks into the decoder module enhances our model’s capacity for recalibrating features, resulting in discernibly improved accuracy in the context of segmentation tasks.

The empirical evaluation of our proposed MetaSwin model, conducted on distinct medical datasets encompassing the BraTS 2023 segmentation challenge dataset and the BTCV dataset, provides compelling evidence of its efficacy and efficiency. we rigorously tested our model using a five-fold cross-validation approach on the BraTS dataset. This method allowed us to thoroughly assess the model’s performance while considering the limited availability of dataset resources. While we did not employ a separate, completely independent test set in this study, we acknowledge the importance of setting aside a portion of the data for such a purpose in future experiments. This added measure will serve to enhance the overall strength and resilience of our model’s performance evaluation, addressing the concern of overfitting more effectively. Notably, the evaluation on the BTCV dataset reveals a noticeable decrease in the test scores of MetaSwin+SE and SwinUNETR+SE. To understand the underlying reasons for this notable decrease when compared to the consistent performance of models without SE blocks, more in-depth analysis and investigation are imperative. Further research is required to elucidate the precise mechanisms responsible for this observed contrast.

The outcomes of our experimentation underscore the practicality and potential of our novel approach, as evidenced by its competitive performance in semantic segmentation, achieved alongside a notable reduction in computational complexity and training duration. This research not only contributes to the ongoing expansion of efficient deep learning architectures, but it also underscores the pivotal role played by thoughtful architectural design choices in yielding remarkable outcomes. By illuminating the capabilities of MetaSwin as a robust and agile tool, we provide valuable insights that facilitate for future advancements and innovations in the realm of semantic segmentation methodologies.

Additional Information and Declarations

Competing Interests

Author Contributions

Data Availability

The authors declare there are no competing interests.

Soyeon Lee conceived and designed the experiments, performed the experiments, analyzed the data, performed the computation work, prepared figures and/or tables, authored or reviewed drafts of the article, and approved the final draft.

Minhyeok Lee conceived and designed the experiments, authored or reviewed drafts of the article, and approved the final draft.

The following information was supplied regarding data availability:

The code is available at GitHub and Zenodo:

- https://github.com/soyeon1608/MetaSwin

- soyeon1608. (2023). soyeon1608/MetaSwin: 1.0.0 (1.0.0). Zenodo. https://doi.org/10.5281/zenodo.10039976.

The BraTS and BTCV datasets are available at https://www.synapse.org/#!Synapse:syn3376386.

The datasets are also available on figshare:

- Lee, Soyeon (2023). BraTS2023 challenge raw data_vol.1. figshare. Dataset. https://doi.org/10.6084/m9.figshare.24440491.v1.

- Lee, Soyeon (2023). BraTS2023 challenge raw data_vol.2. figshare. Dataset. https://doi.org/10.6084/m9.figshare.24440860.v1.

- Lee, Soyeon (2023). BraTS2023 challenge raw data_vol.3. figshare. Dataset. https://doi.org/10.6084/m9.figshare.24441040.v1.

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
