# Peer review of "MetaSwin: a unified meta vision transformer model for medical image segmentation"

_PeerJ Computer Science, doi:10.7717/peerj-cs.1762_

## Round 0.1 · original submission · Major Revisions

The reviewers have substantial concerns about this manuscript. The authors should provide point-to-point responses to address all the concerns and provide a revised manuscript with the revised parts being marked in different color.

**Language Note:** The review process has identified that the English language must be improved. PeerJ can provide language editing services - please contact us at copyediting@peerj.com for pricing (be sure to provide your manuscript number and title). Alternatively, you should make your own arrangements to improve the language quality and provide details in your response letter. – PeerJ Staff

Reviewer 1 ·

Basic reporting

The English language should be improved to ensure that an international audience can clearly understand your text


Line 146 - line 148 please cite related artiles

Line 195: recent research -> recent researches

Line 195: Particular the MetaFormer Study: please include citation

Line 196: such as attention module -> such as the attention module

Line 321: which is -> in which

Figure 1: MetaSwin is the model without only SE blocks. -> MetaSwin is the model without any SE blocks.

Experimental design

The experiment was well designed, 80-20 rule was adopted for the traininng and validation. And a comparison was made between the proposed model MetaSwin and existing models SwinUNETR and SwinUNETR+SE,

On the table results, it’s better to bold the highest score.

Validity of the findings

This paper proporsed a novel medical image segmentation model by integrating the ideas behind Swin Transformer and U-Net, the emphasization of the new model is achieving the comparable results with reduced computational complexity.

Cite this review as

Reviewer 2 ·

Basic reporting

This work provides a comprehensive overview of the evolution of medical image segmentation, with a keen focus on the development of U-NET and the integration of transformer-based models. The study astutely identifies the limitations of SwinUNETR and introduces an innovative approach by adopting Metaformer’s concept of substituting attention blocks with average pooling layers, effectively reducing computational overhead without compromising performance. Moreover, the incorporation of the Squeeze-and-Excitation (SE) Block adeptly captures essential global information in a dynamic manner. The results unequivocally demonstrate the efficacy of these novel methods, offering valuable insights poised to propel future advancements and innovations in the field of semantic segmentation methodologies

Experimental design

For BraTS dataset, the author take a pragmatic approach to model evaluation in light of the absence of a publicly accessible test set. The subsequent implementation of five-fold cross-validation on the data showcases a robust effort to validate the model's performance across diverse subsets. While a separate test set is not utilized, this approach still offers a thorough examination of the model's capabilities. Nevertheless, it's worth considering the potential benefits of reserving a portion of the data for a completely independent test set in future experiments, as it can provide an additional layer of confidence in the model's performance to avoid overfitting issues.

Validity of the findings

1. In Table 1, the author states that MetaSwin achieved the highest average Dice score in Fold 3. However, upon careful examination, it appears to have the lowest score among all four methods. It's essential to verify the results for accuracy. Notably, MetaSwin exhibits the best average Dice score in Fold 2 instead.

2. Figure 2 reveals a significant drop in the test Dice scores for MetaSwin, both in terms of the number of parameters (around 1000) and training time (around 800). To provide a clearer understanding of this phenomenon, it would be beneficial to offer a plausible explanation for the observed contrast compared to the stable curve of MetaSwin-SE.

3. It is recommended to include a legend detailing the colors in Figure 3 for improved clarity and comprehension.

Additional comments

Please consider including a Readme file along with the package version in the GitHub repository. This will greatly facilitate the replication of the experiment

Cite this review as

Reviewer 3 ·

Basic reporting

The study of "MetaSwin: a unified meta vision transformer model for medical image segmentation" proposed a framework of MetaSwin plus SE for medical image segmentation by replacing attention module with general pooling operator and adding the SE block. However, there are some items need to be revised and improved.

Experimental design

Experiments: The manuscript performed two different comparisons for two different dataset. BraTS shows the Five-fold cross-validation performance, and BTCV shows results of different sizes when using different models. Single experiment can’t support the conclusion well. And why is there only one data point for each model in Figure 4?

Validity of the findings

Results: The average Dice Score in Table 1 from MetaSwin+SE (0.55927) is worse than that of SwinUNETR+SE (0.56554); and the highest accuracy from MetaSwin+SE (81.00%) is also worse than that of SwinUNETR+SE (81.02%). Results show worse performance for proposed model.

Additional comments

Novelty: There was the PoolFormer proposed in MetaFormer article as an example to replace attention module in transformer model with the pooling operator. It uses much fewer (35% - 61%) paramerters and MACs. However, this manuscript also just replace the attention module with the pooling operator, and there’s no other alternatives tested for comparison.

Cite this review as

---

## Round 0.2 · accepted · Accept

Reviewers are satisfied with the revisions, and I concur to recommend accepting this manuscript.

Reviewer 2 ·

Basic reporting

The authors has satisfactorily addressed the concerns and questions previously raised, and revised the manuscript accordingly. At this point, I have no further comments and the paper could be accepted for publication.

Experimental design

I think author have give a detail response against my question towards experimental design.

Validity of the findings

The additional revised manuscript have provide more detailed explanation towards the results, which make the paper more convincing.

Cite this review as

Reviewer 3 ·

Basic reporting

All my concerns have been well addressed. The manuscript is ready to be published.

Experimental design

The study has detailed and reasonable experimental design.

Validity of the findings

Results are correct and clear.

Cite this review as